# Focused clamping of a single neuronal SNARE complex by complexin under high mechanical tension

Min Ju Shon [ID] [1], Haesoo Kim[1] & Tae-Young Yoon [ID] [1]

Neuronal soluble *N*-ethylmaleimide-sensitive factor attachment protein receptors (SNAREs) catalyze synaptic vesicle fusion with presynaptic membranes through the formation of SNARE complexes. Complexin (Cpx) is the only presynaptic protein that tightly binds to SNAREs and regulates membrane fusion, but how it modulates the energy landscape of SNARE complex assembly, especially under mechanical tension on the complex, remains unclear. Here, using magnetic tweezers, we report how Cpx interacts with single SNARE complexes. The effects of Cpx manifest only under high mechanical tensions above 13 pN. Cpx stabilizes the central four-helix bundle of SNARE motifs and, at the same time, prevents the complete zippering of SNAREs by inhibiting linker-domain assembly. These results suggest that Cpx generates a focused clamp for the neuronal SNARE complex in a linker-open conformation. Our results provide a hint as to how Cpx cooperates with neuronal SNAREs to prime synaptic vesicles in preparation for synchronous neurotransmitter release.

[1] School of Biological Sciences and Institute for Molecular Biology and Genetics, Seoul National University, Seoul 08826, South Korea. These authors contributed equally: Min Ju Shon, Haesoo Kim. Correspondence and requests for materials should be addressed to T.-Y.Y. (email: tyyoon@snu.ac.kr)

Exocytosis of synaptic vesicles is tightly regulated to permit robust neurotransmission. To elicit fast synchronous neurotransmitter release, a pool of vesicles docked to the presynaptic plasma membrane fuse with the membrane upon $Ca^{2+}$ influx[1]. Neuronal soluble *N*-ethylmaleimide-sensitive factor attachment protein receptor (SNARE) complexes constitute the core engine of the membrane fusion machinery[2], in which the zippering of SNARE proteins delivers the large energy required to overcome the energy barrier against fusion. Additionally, to inhibit the fusion in resting period and to complete it rapidly within a millisecond when triggered, regulatory proteins, such as complexin (Cpx) and synaptotagmin provide essential points of control[3].

Cpx is the only presynaptic protein that is known to stably bind to the neuronal SNARE complex[4], suggesting that it may directly regulate the assembly process of neuronal SNAREs. Research over the last two decades into the role Cpx plays in neurotransmission has produced a variety of proposals as to its function (see refs. [5,6] for recent reviews). The interpretation of some of the earliest experimental results was complicated by the different effects of Cpx on evoked and spontaneous synaptic vesicle releases[5,7]. Cpx generally facilitates $Ca^{2+}$-triggered release[8], but blocks spontaneous fusion[9,10]. For example, experimental results with invertebrate neurons, such as the neuromuscular junctions of *Drosophila* highlights the clamping effect of Cpx on spontaneous fusion[10], which is ascribed to the highly charged accessory helix of *Drosophila* Cpx[11]. However, such clamping effect diminishes when mammalian Cpx, which contains less charges on the accessory helix, replaces the endogenous Cpxs in *Drosophila*[12,13]. The mechanistic details of how Cpx influences a SNARE complex therefore remain controversial[11,14–16], partially due to the inconsistent preparations of truncated Cpx variants and SNARE proteins and different biological systems used across experiments.

These effects of Cpx have been attributed to the way its multiple domains interact with the SNARE complex[17,18]. Cpx is a short linear protein that consists of short helical domains and unstructured regions. The central helix of Cpx is primarily responsible for the strong binding of Cpx to the neuronal SNARE complex[19–21], and its presence is essential in observing virtually all of the known effects of Cpx. The N-terminal domain and accessory helix also associate with neuronal SNAREs, affecting vesicle fusion in distinct ways. The N-terminal domain that potentially forms a short amphipathic helix is reported to interact with both the SNARE complex and the plasma membrane, facilitating fusion in general[17,22–24]. In contrast, the accessory helix has been frequently hypothesized to interfere with the SNARE complex assembly and inhibit fusion[22,25–28]. Lastly, the C-terminal domain of Cpx is less structured than the helical segments and reported to guide synaptic vesicles to fusion sites[29,30].

In a physiological milieu, however, all these domains of Cpx would work in a concerted manner, requiring the function of Cpx to be understood as a whole protein. Furthermore, how mechanical tension in the neuronal SNARE complexes affects the reported effects of Cpx has not been explored so far. It has been presumed that the tension rapidly builds up when a synaptic vesicle approaches a presynaptic membrane because of the electrostatic repulsion, hydration barrier, and steric hindrance between the two fusing membranes[31,32]. This tension in a SNARE complex was shown to significantly tilt the energy landscape that governs SNARE zippering processes[33,34]. The mechanical tension may also radically reshape the ensuing interactions between Cpx and SNAREs. Unfortunately, it remains unknown how the different parts of Cpx work together on a single SNARE complex under such force-loaded conditions. In particular, while Cpx has

been suggested to clamp partially zippered SNARE complexes and thus prevent membrane fusion[21,25–28], it has proven difficult to observe such an intermediate conformation of the neuronal SNARE complex directly, likely because of the transient nature of this state in the absence of applied tension.

In this work, we studied how mammalian Cpx modulates the conformation of the neuronal SNARE complex under mechanical tension. While applying 12–16 pN of tension using magnetic tweezers, we directly monitored the rapid transition between intermediate conformations of single neuronal SNARE complexes. By adding Cpx to the mechanical unzipping and rezipping cycles of the pre-assembled SNARE complexes, we found that Cpx significantly stabilizes the assembled SNARE complexes. At the same time, Cpx blocks complete zippering of the SNARE complex by interfering with assembly of the SNARE linker domains. The stabilizing and the inhibitory effects are distinctly mediated by the central-accessory helices and the N-terminal domain of Cpx, respectively. Overall, the two effects of Cpx jointly promote a "linker-open" state of the SNARE complex, thereby clamping the complex in a focused, partially zippered conformation. Importantly, both of the molecular effects of Cpx manifest in a narrow range of applied forces, namely 13–16 pN, implying that Cpx might be naturally tuned to function under a well-defined range of mechanical tensions applied to individual SNARE complexes.

## Results

**Manipulation and observation of single SNARE complexes**. We employed magnetic tweezers to study the conformations of single neuronal SNARE complexes during mechanical manipulation[33] (Fig. 1a,b). We pre-assembled ternary SNARE complexes consisting of syntaxin-1A, SNAP-25, and synaptobrevin-2, and attached two 510-bp DNA handles (Fig. 1b and Supplementary Fig. 1). The DNA handles were then bound to a glass surface and to a magnetic bead, which collectively formed a pulling construct (Fig. 1a and Supplementary Table 1). Because the DNA handles were attached to the C-terminal ends of synaptobrevin-2 and syntaxin-1A via two artificial cysteine residues[35], tension was generated from the C-terminus of the SNAREs, mimicking the force-loaded environment assumed for neuronal SNARE complexes during synaptic vesicle fusion[36] (Fig. 1a,b and Supplementary Fig. 2; transmembrane domains were truncated). To enable multiple cycles of interrogation, we covalently linked the N-termini of synaptobrevin-2 and syntaxin-1A by introducing two additional cysteines that formed a disulfide bond (Fig. 1b). This N-terminal knotting also allowed study of only properly folded SNARE complexes during our tweezing experiments. Misfolded SNARE complexes such as anti-parallel SNAREs failed to form the disulfide crosslinking, ruptured under high mechanical tension, and were subsequently excluded from our observations. As we varied the force applied to the magnetic bead by moving the permanent magnets, we tracked the vertical position of the bead at 100 Hz to monitor the conformation of a single SNARE complex. For the studies of Cpx function, we later introduced Cpx proteins to the assay through microfluidic buffer exchange.

When we increased the magnetic force at a loading rate of 1 $pN\,s^{-1}$, the SNARE–DNA construct displayed a characteristic force–extension curve for a 1-kbp double-stranded DNA, consistent with the length of the two DNA handles (510 bp each) (Fig. 1c). When the tension was increased over 13 pN, we observed an abrupt upward movement of the bead by ~25 nm, which indicated mechanical unzipping of the SNARE complex (Fig. 1d, black arrow). Our modeling suggested that this 25-nm unzipping corresponds to the full unraveling of synaptobrevin-2

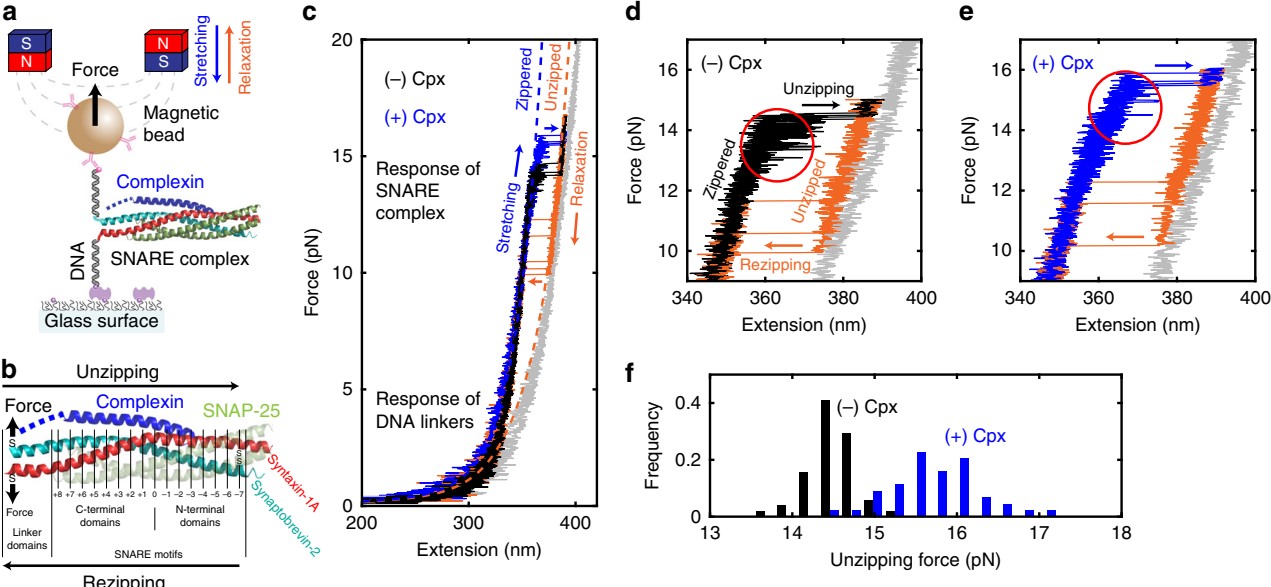

**Fig. 1** Mechanical stabilization of SNARE complexes by complexin. **a** SNARE–DNA hybrid construct in magnetic tweezers setup. **b** A predicted three-dimensional structure of the neuronal SNARE complex bound to a molecule of Cpx (PDB 3IPD[49] aligned to 1KIL[20]). The N-terminal domain of Cpx not determined in the crystal structure (1–25) is represented with a blue dashed line. The letters "S" denote the thiol groups of artificial cysteine residues. The numbers below the structure indicate the leucine-zipper layers. **c** Typical force–extension curves from a single SNARE complex in the absence (black) and presence (blue) of Cpx. The black and blue traces show the stretching period that features "unzipping" events from the fully zippered to the unzipped state, while the orange traces show the relaxation period with the opposite "rezipping" events. In gray, a curve for the unfolded SNAREs that fail to re-zip is shown. Dashed lines indicate the models of extension for the corresponding conformations of SNARE complex (Supplementary Note 3). **d**, **e** Force–extension curves from a single SNARE complex in the absence (**d**) or presence (**e**, same graph as in **c**) of Cpx. **f** Unzipping force distributions from a single SNARE complex in the absence (black, $N = 51$) and presence (blue, $N = 44$) of 5 μM Cpx

from the rest three-helix bundle composed of syntaxin-1A and SNAP-25 (Supplementary Fig. 3 and Supplementary Note 3). We also observed appreciable, reversible movements of the bead preceding the main unzipping event, implying the existence of intermediate states for the tweezed SNARE complex[33,37–39] (Fig. 1d, red circle).

Shortly after the unzipping event, we lowered the magnetic force to record the force–extension curve during relaxation (Fig. 1c, d, orange traces). We observed backward transitions of comparable size to the unzipping event (Fig. 1c, d, orange arrows). We reasoned that this reverse transition reflects the full re-assembly, or "rezipping", of the SNARE complex because the force–extension curve snapped back exactly to that observed during stretching. The rezipping forces lower than unzipping indicated a mechanical hysteresis in the force–extension cycle of a single SNARE complex[33].

Finally, when we increased the force beyond the unraveling of synaptobrevin-2 (>15 pN), we observed an additional upward movement of the bead by ~5 nm (Fig. 1c, d, transition to gray trace) due to the dissociation of SNAP-25. Addition of free SNAP-25 proteins at 2.5 μM to the assay buffer allowed re-association of SNAP-25 with the pulling construct in the low-pN region[34] (Fig. 1c, gray to black).

**Cpx mechanically stabilizes neuronal SNARE complexes.** Next, we introduced 5 μM full-length Cpx (rat Cpx-1) to the SNARE complex being tweezed (Fig. 1c, e). Because the dissociation constant between Cpx and a SNARE complex is known to be 10–70 nM[40–42], 5 μM was sufficiently high such that all SNARE complexes were bound to Cpx. Remarkably, addition of Cpx caused the unzipping to occur at higher levels of force (Fig. 1c, e, blue arrows). Across different force-loading rates, we consistently

observed this Cpx-dependent resistance of SNARE complex to high forces (Supplementary Fig. 4). Moreover, when we looked closely at the force–extension curves, Cpx was found to substantially suppress the reversible transitions before unzipping (Fig. 1e versus 1d, red circles), implying that Cpx influences the conformation of the tweezed SNARE complexes before unzipping.

From repeated stretching and relaxation cycles, we collected the force levels at which the unzipping events occurred (Fig. 1f). The unzipping force varied randomly across the trials, indicating stochastic crossings of an energy barrier. Comparing the distributions of the unzipping force showed that in the presence of Cpx, the unzipping occurred at higher force levels by ~2 pN on average (Fig. 1f). Interestingly, the broader distribution of unzipping force in the presence of Cpx implied more heterogeneous rupture events at the single-molecule level. The rezipping force, in contrast, did not change significantly upon the addition of Cpx (Supplementary Fig. 4). Collectively, our results suggest that Cpx significantly enhances the mechanical stability of the neuronal SNARE complex.

**Cpx extends the lifetimes of zippered SNAREs under tension.** To quantitatively assess the stabilization of SNARE complex by Cpx, we carried out force-jump experiments and measured the lifetimes of zippered and unzipped SNARE complexes under tension (Fig. 2). The use of the magnetic tweezers enabled swift changes and stable maintenance of force levels with simple instrumentation. We first increased the load on a fully assembled SNARE complex from 10 to 14 pN rapidly within 100 ms, and then measured the latency to unzipping, $\tau_{unzip}$ (Fig. 2a). We then reversed the force scheme and abruptly lowered the force from 14 to 10 pN to measure the latency in rezipping, $\tau_{rezip}$.

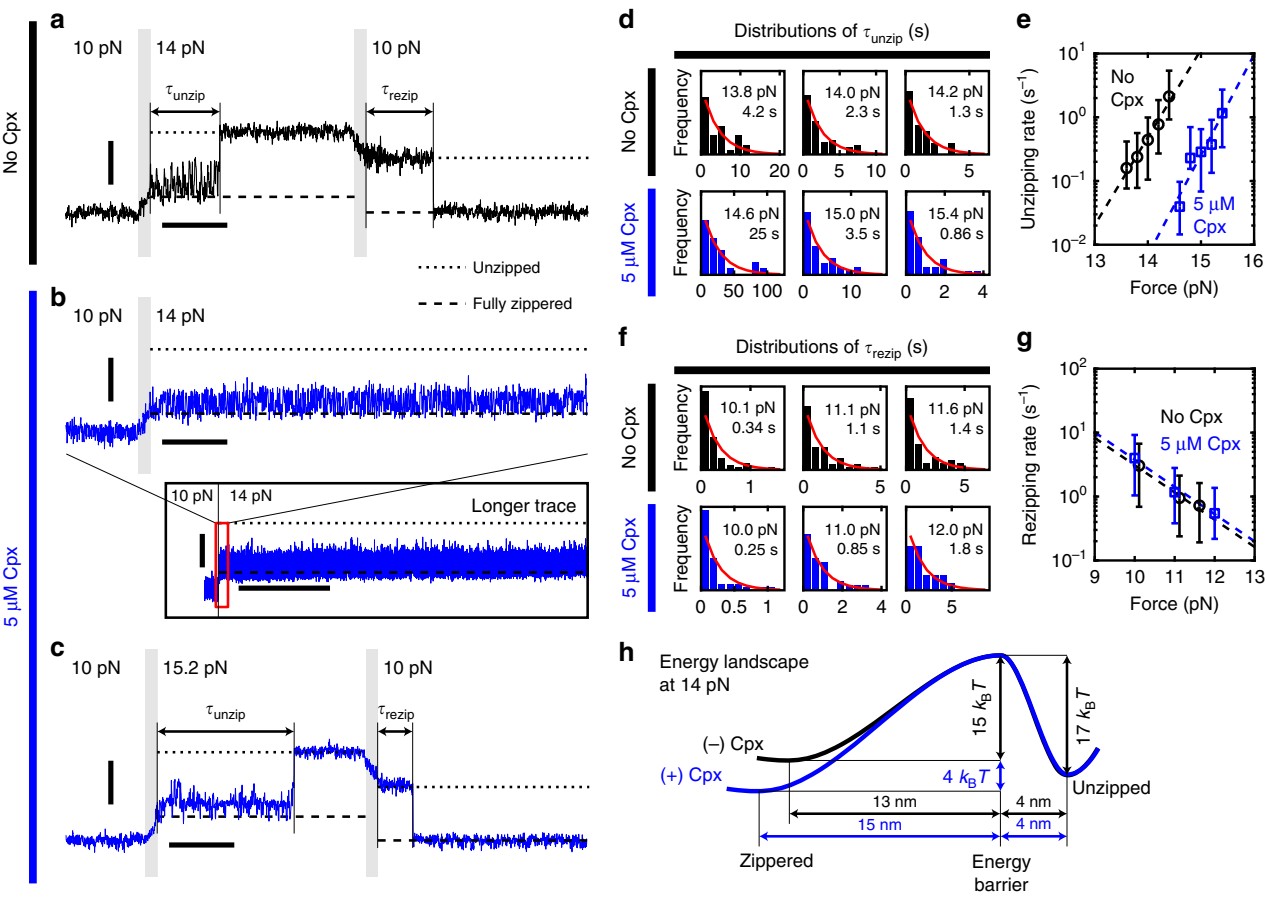

**Fig. 2** Complexin extends the lifetime of zipped SNARE complexes under tension. **a–c** The force-jump scheme used for serial measurements of $\tau_{unzip}$ and $\tau_{rezip}$ in the absence (**a**) or presence (**b**, **c**) of Cpx. The predicted locations of the fully zippered and the unzipped states are indicated with dashed and dotted lines, respectively. Gray vertical bars mark the force-jump periods. Inset in **b** shows the marked increase in lifetime of SNARE induced by Cpx, the red box of which is magnified above. Vertical scale bars represent 20 nm, and horizontal scale bars represent 0.5 s (main panels) and 30 s (inset in **b**). **d**, **f** Distributions of $\tau_{unzip}$ (**d**) and $\tau_{rezip}$ (**f**) at varying forces ($N \geq 19$ for each panel). Red curves represent the fits to exponential distributions with their mean values indicated. **e**, **g** Force dependences of unzipping (**e**) and rezipping (**g**) rates in the absence (black) or presence (blue) of Cpx. Error bars represent the 95% confidence intervals for the mean parameter estimates in **d** and **f**. Dashed lines represent the fits to the Bell equation. **h** The energy diagram for the unzipping/zippering of a neuronal SNARE complex at 14 pN. The effect of Cpx is indicated in blue

Without Cpx, the SNARE complexes mostly unzipped within a few seconds at 14 pN (Fig. 2a). In contrast, addition of 5 μM Cpx dramatically extended $\tau_{unzip}$ to hundreds of seconds (Fig. 2b). The unzipping still occurred in the presence of Cpx albeit at forces above 14 pN, and applying ~15 pN shortened $\tau_{unzip}$ to an experimentally accessible timescale (Fig. 2c). All the distributions of $\tau_{unzip}$ obtained at different force levels followed single-exponential distributions (Fig. 2d), indicating the existence of a single major energy barrier to unzipping. Note that the logarithms of the unzipping rates depended linearly on the applied force (Fig. 2e), confirming the validity of the Bell equation even under the molecular actions of Cpx[43,44] (Supplementary Note 4). Comparing the unzipping rates showed that Cpx slowed down the unzipping of SNARE complexes by nearly two orders of magnitude across the force range we investigated (Fig. 2e).

The observed $\tau_{rezip}$ values also followed single exponential distributions, yielding SNARE zippering rates at varying forces (Fig. 2f). The logarithms of the rezipping rates decreased linearly with the applied force again as predicted by the Bell equation (Fig. 2g). However, the presence of Cpx did not appreciably change $\tau_{rezip}$, consistent with the invariance of rezipping force distribution (Supplementary Fig. 4).

From the linear dependences of the logarithms of the kinetic rates on the applied mechanical tension (Fig. 2e, g), we deduced

distances and heights of the energy barriers crossed during unzipping and rezipping of a single neuronal SNARE complex. We present a coarse-grained sketch of this energy landscape in the presence or absence of Cpx (Fig. 2h; we used a pre-exponential factor of $10^6 \, s^{-1}$ as generally accepted[33,34]). In this energy landscape, the zippered state refers to a collection of all conformations before the main unzipping, including the fully assembled and partially unzipped forms (see Fig. 3 below). With force, the zippered state becomes less stable and energetically comparable to the unzipped state (Fig. 2h). The estimated height of the energy barrier to unzipping was 15 $k_BT$. Importantly, Cpx stabilized the zippered state by 4.3 ± 0.5 $k_BT$ (error: 95% CI) and consequently raised the barrier for unzipping to 19 $k_BT$, accounting for the extended lifetimes of zippered SNARE complexes (Fig. 2h, blue versus black curves). The unzipped state, in contrast, was unaffected by the presence of Cpx. Together, our results suggest Cpx markedly affects the energy landscape governing assembly and disassembly of the neuronal SNARE complex.

**Cpx affects unzipping intermediates of a SNARE complex.** So far, we have focused on the mechanical stabilization of SNARE complexes by Cpx. However, there were two observations that could not be explained by this stabilizing effect. First, Cpx shifted

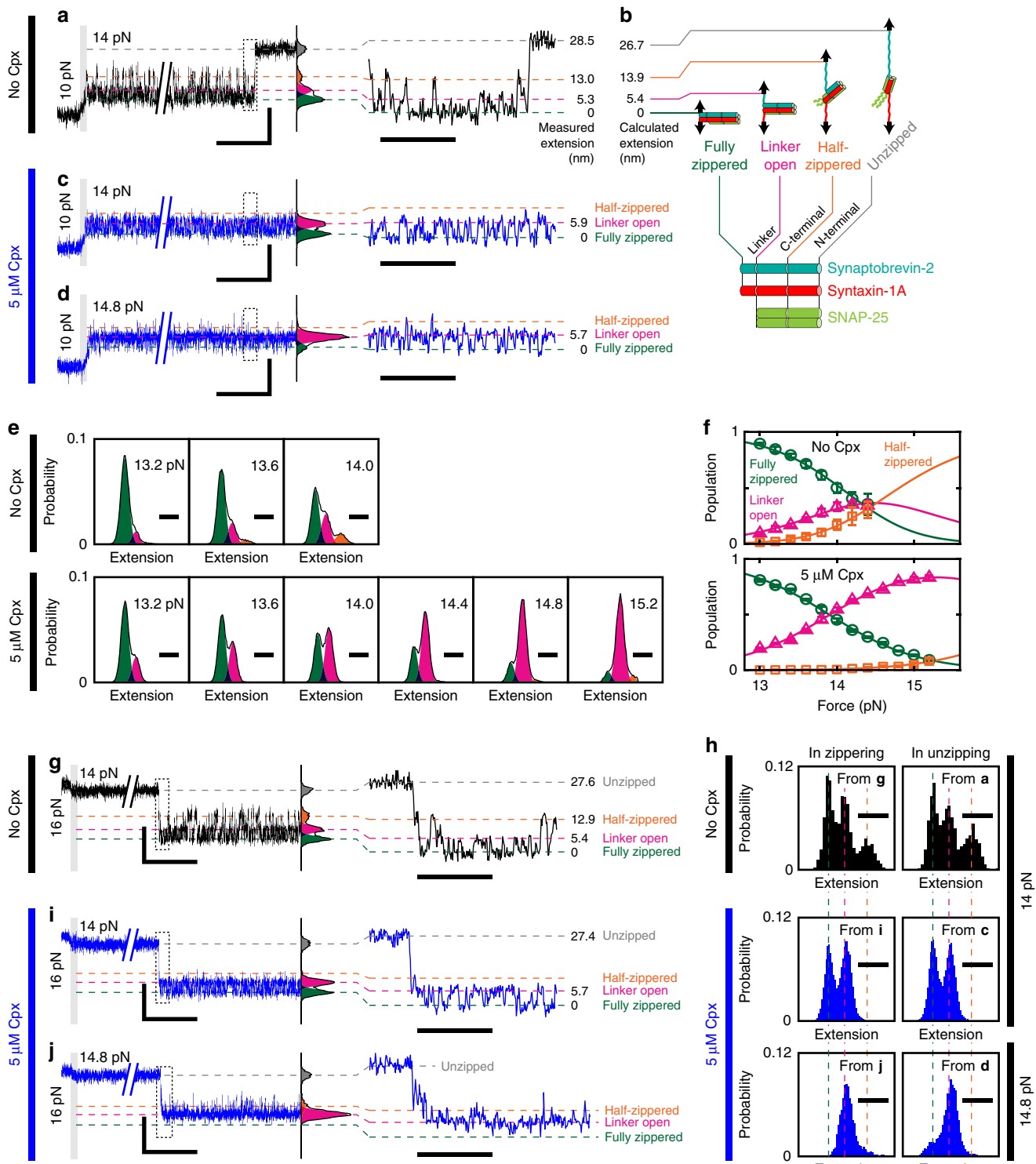

**Fig. 3** Complexin promotes the linker-open state of neuronal SNARE complexes. **a**, **c**, **d** High-speed observations (1.2 kHz) of the conformational intermediates during the unzipping of a SNARE complex. The measured distributions of extensions (black lines displayed to the right of the traces) are overlaid with their identified Gaussian components color-coded green, magenta, orange, and gray from the lowest. Dotted areas are magnified on the right. The locations of the peaks are indicated with dashed lines of the same color and with numbers. **b** A model for the intermediate conformations of a SNARE complex with calculated values of extension. **e** Force-dependent shifts in extension distribution with the indicated Gaussian components. The color code is the same as in **a**–**d**. **f** Force-dependent equilibria among the three zippered conformations in the absence (upper) and presence (lower) of Cpx. Solid lines represent the fits to a three-state equilibrium model (Supplementary Note 5). Data are represented as mean ± s.d. of more than three measurements. **g**, **i**, **j** High-speed observations of the conformational intermediates during the zippering of a SNARE complex. The color code is the same as in **a**–**f**. **h** Comparisons of conformational equilibria during unzipping (upper; 10 → 14 pN) and zippering (lower; 16 → 14 pN) of a single SNARE complex. In all panels, vertical scale bars represent 20 nm. Horizontal scale bars represent 1 and 0.1 s (on left and right in **a**, **c**, **d**, **g**, **i**, **j**, respectively), and 10 nm (in **e**, **h**)

the coarse-grained zippered state of SNARE complex farther from the unzipped state by about 2 nm (Fig. 2h). Second, in the force-ramp experiments in Fig. 1, we observed that Cpx suppresses the fluctuations in SNARE complex preceding the main unzipping event. These results possibly suggest that Cpx affects the conformation of SNAREs before the unzipping takes place[40].

We therefore sought to resolve the intermediate conformations of single SNARE complexes at a higher sampling rate of 1.2 kHz (Fig. 3a). First, we repeated our force-jump experiments in Fig. 2a, increasing the tension from 10 to 14 pN to induce SNARE complex unzipping. Without Cpx, the extension values of a single SNARE complex sampled at 14 pN formed a mixture of Gaussian distributions (Fig. 3a). Notably, these components were not resolved by our previous 100-Hz tracking of the beads, indicating the importance of the high sampling rate (Supplementary Fig. 5).

We assigned the lowest peak to the fully zippered SNARE complex and defined it as 0 nm for simplicity. The next two peaks, located at 5.3 and 13.0 nm, matched the expected lengths for the linker-open (5.4 nm) and the half-zippered state (13.9 nm), respectively (Fig. 3b versus 3a; see Supplementary Note 3 for calculations of the expected lengths). For the linker-open conformation, we assumed fully folded four-helix bundles of SNARE motifs and random-coil linker domains void of secondary structures (Fig. 3b)[34,45]. For the half-zippered state, we included unzipping of synaptobrevin-2 and the Q-SNARE proteins up to the +2 and +4 layers, respectively[11,37,46]. The peak after the main unzipping was placed at 28.5 nm. This extension best agreed with complete unraveling of synaptobrevin-2 and further unfolding of the Q-SNARE proteins to the zeroth layer, which was estimated to be at 26.7 nm (Fig. 3b versus 3a and Supplementary Fig. 3).

Remarkably, when we added 5 μM Cpx, the extension distribution observed under 14 pN changed substantially. The population of the linker-open state increased at the expense of the fully zippered and half-zippered states (Fig. 3c versus 3a). To determine whether this finding was a peculiar feature at 14 pN, we repeated the experiment at a higher force level of 14.8 pN. Here, the linker-open population further increased while the other conformations were only transiently adopted, reaffirming the promotion of the linker-open state by Cpx (Fig. 3d).

To systematically test this trend, we investigated the intermediate populations while gradually increasing the applied tension (Fig. 3e, f, and Supplementary Note 5). In the absence of Cpx, at least 13 pN was required to induce any partial unzipping of the SNARE complex. Above 13 pN, both the linker-open and half-zippered states were populated with increasing tension (Fig. 3e, upper panels). In the presence of Cpx, the unzipped intermediates became observable again at 13 pN, thus marking the same threshold force level as that observed without Cpx (Fig. 3e, f). Immediately above 13 pN, however, the linker-open population was substantially promoted with virtually no increase in the half-zippered state, leaving the linker-open conformation as the only abundant species at forces above 14.5 pN (Fig. 3e, lower panels). The distance between the fully zippered and the linker-open state was unaffected by Cpx (Supplementary Fig. 6), implying that the linker-open conformation was largely conserved.

**Linker-open state is promoted in unzipping and zippering**. Because the promotion of linker-open state by Cpx was observed in the unzipping processes, we wondered whether the same effects would be reproduced during a zippering process. To initiate zippering at 14 pN, we first induced full unzipping of a SNARE complex at 16 pN, lowered the tension to 14 pN, and waited until a rezipping event took place (Fig. 3g). Above 13 pN, the zippering

reaction was not only slow but also competed by the dissociation of SNAP-25, especially without Cpx. Nevertheless, we were able to detect such zippering events at or above 14 pN. When successful, the SNARE complex rezipped after a long latency and exhibited transitions between intermediates (Fig. 3g). Of note, the distribution of extension exhibited by this rezipped SNARE complex was nearly the same as that observed during the unzipping process (Fig. 3g versus 3a, and Fig. 3h).

Next, we added 5 μM Cpx to this force-jump scheme for SNARE rezipping (Fig. 3i). Importantly, we found the effects of Cpx to appear immediately after rezipping within our time resolution (~830 μs), exhibiting an enhanced linker-open population while repressing the half-zippered state. We further confirmed that the linker-open state was the only prevailing conformation at 14.8 pN of tension (Fig. 3j). This rezipping experiment essentially recapitulated the promotion of linker-open state through the suppression of the other states. Indeed, the conformational distributions observed during the unzipping and rezipping experiments matched well with one another (Fig. 3h), suggesting that the fast exchanges among the intermediates occur near an equilibrium. In other words, the relative populations were determined solely by the moment-to-moment tension regardless of how we applied the force.

Taken together, two seemingly paradoxical effects of Cpx became evident. On one hand, Cpx nearly abolished the half-zippered population across the entire force range we studied (Fig. 3f, orange), consistent with the stabilizing role of Cpx described in Figs. 1 and 2. On the other hand, Cpx preferentially increased the linker-open population, preventing the SNARE complex assembly from reaching completion (Fig. 3f, magenta). The first effect assists the zippering of SNARE motifs while the second effect conversely leads to an incomplete zippering of the linker regions. These two actions of Cpx collectively induced a focused clamping of single neuronal SNARE complexes in the linker-open conformation.

**Kinetics of Cpx-mediated linker-open SNARE complex**. To understand the kinetics of exchanges among the conformational intermediates, we applied hidden Markov modeling (HMM) to the high-speed traces recorded with or without Cpx (Fig. 4a and Supplementary Note 6). For the prior inputs in the HMM analysis, we adopted a few parameters from the measured extension distributions for the three conformational intermediates (Fig. 3e), such as the means and variances of the Gaussian distributions representing respective intermediates. The HMM procedure identified all the transitions between intermediates, and thus the corresponding kinetic rates were estimated (Fig. 4a). All the obtained kinetic rates fell in the rage between 1 and $10^3\,s^{-1}$, which could be reliably determined from our 1.2-kHz time traces (Fig. 4b–e).

The HMM analysis showed that the zippering rate of the C-terminal half of SNARE motifs (i.e., transition rate from the half-zippered to the linker-open state) was increased by Cpx (Fig. 4b). Conversely, the unzipping rate for the C-terminal half was substantially reduced by Cpx (Fig. 4c). Thus, Cpx not only promoted the zippering of SNARE motifs' C-terminal half, but it also prevented unzipping of the same region, accounting for the observed stabilization.

Consistent with the observed promotion of the linker-open state, Cpx decelerated the zippering of linker domains (i.e., transition from the linker-open to the fully zippered state) (Fig. 4d). However, the unzipping rate of the linker domains was only marginally affected by Cpx (Fig. 4e), suggesting a rather passive role of Cpx in the inhibition of linker domain assembly. We noted that in the high-force regime we studied, the unzipping effect of the loaded

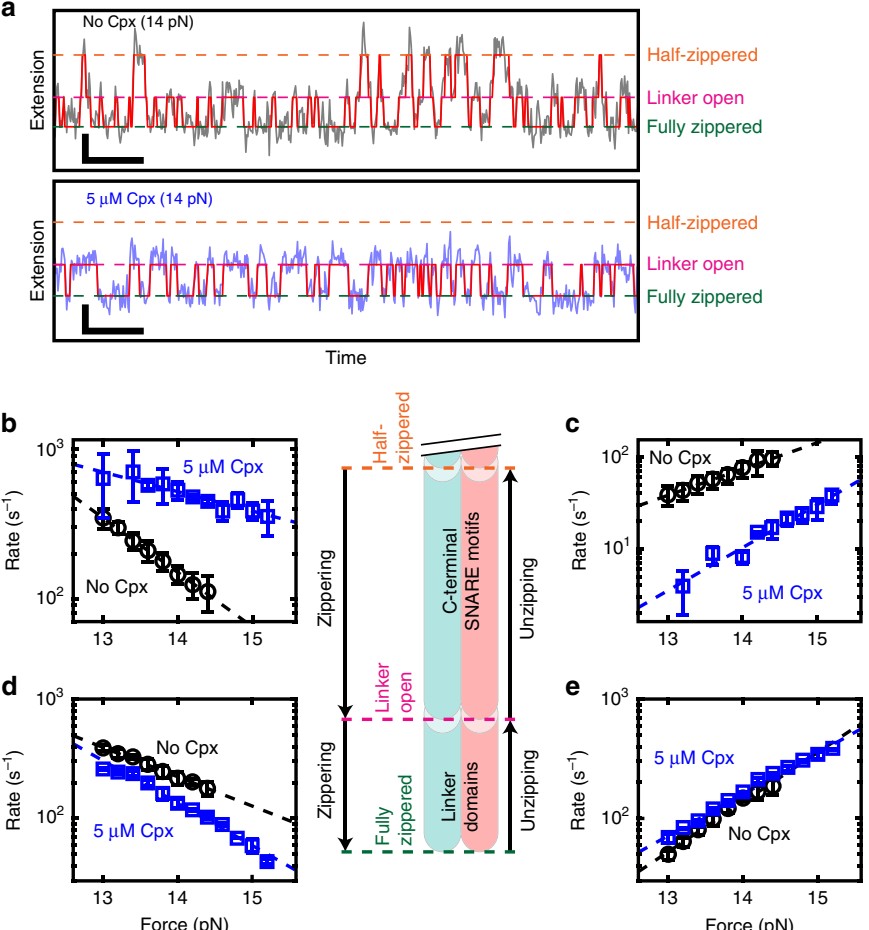

**Fig. 4** Kinetic analysis of conformational intermediates through hidden Markov modeling. **a** Extension time traces measured at 1.2 kHz with a single SNARE complex in the absence (upper) or presence (lower) of 5 μM Cpx. The state paths obtained from hidden Markov modeling are overlaid in red (Supplementary Note 6). Vertical and horizontal scale bars represent 5 nm and 50 ms, respectively. **b–e** Force dependence of zippering (**b**, **d**) and unzipping (**c**, **e**) rates for C-terminal SNARE motifs (**b**, **c**) and linker domains (**d**, **e**) in the absence (black) or presence (blue) of 5 μM Cpx. Data are represented as mean ± s.d. of more than three measurements. Dashed lines represent the fits to the Bell equation. A diagram for the corresponding transitions in a SNARE complex is shown in the center

tension would overwhelm any destabilizing effect of Cpx. On the contrary, extrapolation of Fig. 4e towards 0 pN suggests that the unzipping would become substantially faster in the presence of Cpx albeit on much longer time scales (Supplementary Fig. 7a), implying that Cpx might actively open the linker domains in such a low-force regime. This trend is consistent with a previous single-molecule fluorescence resonance energy transfer (FRET) study that reported lowering of the FRET efficiency at the C-terminal ends of SNAREs by Cpx[40]. Together, these kinetic analyses revealed the details of how Cpx enables the focused clamping of neuronal SNARE complexes in the linker-open state. Finally, from the force dependence of the reaction rates, we obtained an extensive set of thermodynamic parameters that describes the energy landscape across the zippered states, which re-affirmed our conclusion (Supplementary Fig. 7 and Supplementary Table 2).

**Differential regulation of SNARE complex by domains of Cpx.** To determine which domains of Cpx underlie its two-faceted functionality, we prepared truncated Cpx variants that sequentially lack domains from its ends (Fig. 5a and Supplementary Fig. 1). We first examined these variants in force ramps and compared their force–extension curves as done in Fig. 1 (Fig. 5b). We observed an increase in unzipping force for all the Cpx variants, including the central helix-only construct (termed "C").

This result suggests the stabilization largely depends on binding of the central helix to the SNARE complex[19,20].

We next explored the force-dependent equilibrium of the zippered intermediates for the same Cpx variants. We fitted the distributions of their extension values to three Gaussian distributions in order to delineate the evolution of intermediate populations with increasing force (Fig. 5c–f; see Supplementary Fig. 7 and Supplementary Table 2 for relevant HMM analyses). We reconfirmed that the suppression of half-zippered state depended mainly on the central helix of Cpx. For example, the solitary central helix substantially reduced the half-zippered population at 14 pN (Fig. 5e; C versus None). In addition, the accessory helix was essential in extinguishing the remaining half-zippered population (Fig. 5e; AC versus C), suggesting the central and accessory helices work in concert to stabilize the four-helix bundle of SNARE motifs. When examining individual traces, we indeed observed a cumulative suppression of the half-zippered state with sequential addition of the central and accessory helices (Fig. 5g).

We then probed the linker-open population in which wild-type (WT) Cpx clamped the SNARE complex (Fig. 5d, f). Notably, the presence of the N-terminal domain more than tripled the linker-open population at 14 pN, indicating that the ability of Cpx to interfere with linker domains critically depended on its N-terminal domain (Fig. 5f; WT/NAC versus AC). The linker-open

populations for the AC and C variants were, in fact, smaller than that produced in the absence of Cpx (Fig. 5f). This effect was discernible even at the level of individual high-speed traces (Fig. 5h), and clearly illustrates that the stabilizing action of Cpx predominates when its N-terminal domain was missing.

Finally, deletion of the C-terminal domain had no appreciable effect on any of the aspects of Cpx function we studied (Fig. 5b–h; WT versus NAC), consistent with the reported independent role of this domain in membrane binding and localization to the fusion sites[47,48].

## Discussion

Synaptic vesicle fusion, one of the fastest exocytosis processes found in biological systems, is catalyzed by the assembly of neuronal SNARE complexes. Cpx is known to be the only

presynaptic protein that strongly binds with the SNARE complex, raising a possibility that Cpx directly modulates the assembly of SNAREs. Using magnetic tweezers, we observed that Cpx significantly enhances the mechanical stability of single SNARE complexes. Our observation is consistent with the previous deuterium exchange data that Cpx physically protects the four-helix bundle of SNARE motifs[20]. In addition, we found an unexpected role of Cpx in actively driving SNARE complexes into linker-open conformation. Although our observations are based on rat Cpx and SNARE proteins, the results will likely be conserved for human proteins because only five amino acid residues are different (see Methods).

Although the linker domains of SNAREs were known to form a coiled-coil structure extending the helical structure of the SNARE complex up to the transmembrane domains[49], their importance

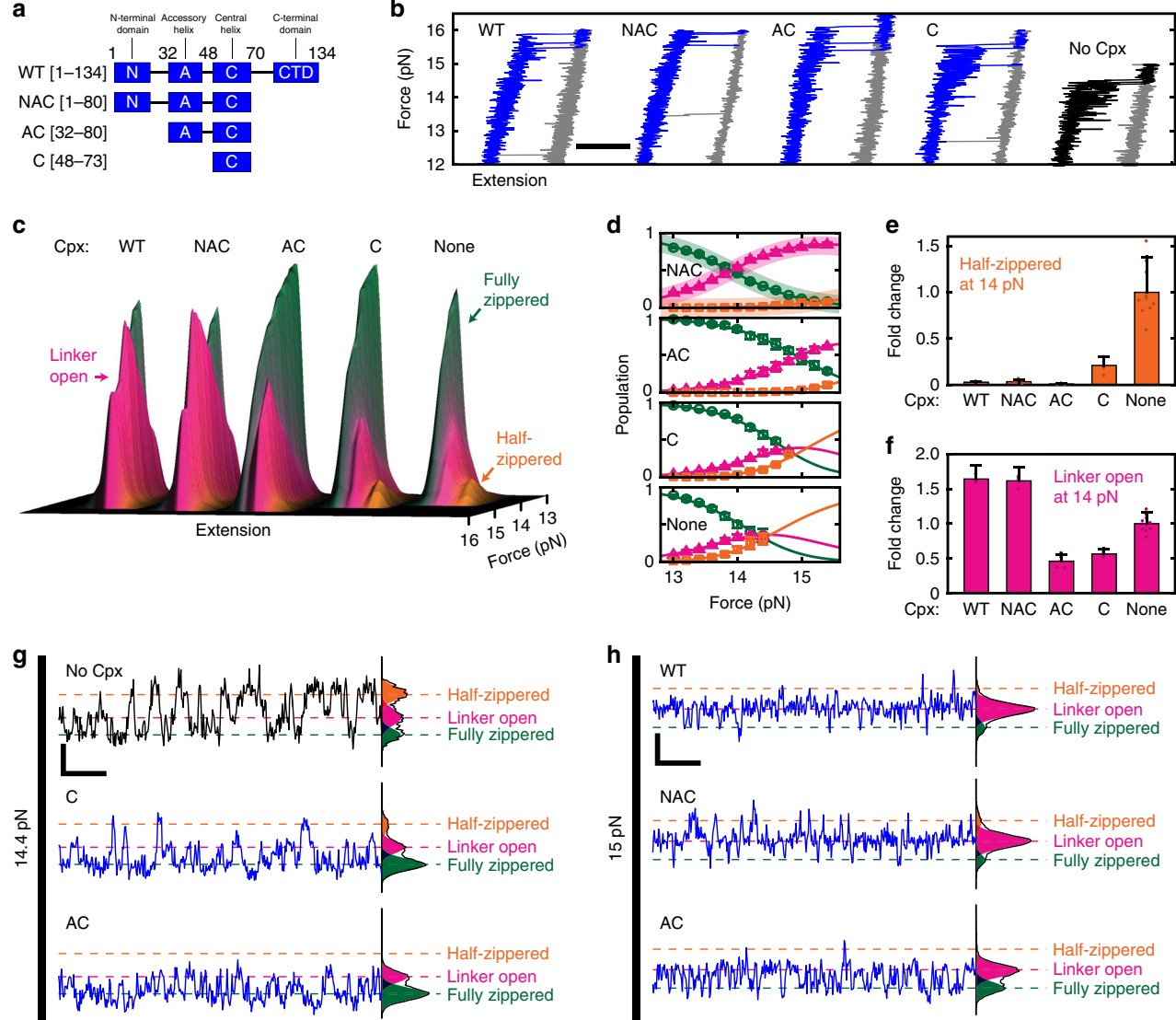

**Fig. 5** Multiple domains of complexin collectively drives SNARE complexes into the linker-open conformation. **a** The domain structures of wild-type Cpx and its variants prepared in this study. **b** Force–extension curves for unzipping (blue/black) and rezipping (gray) of SNARE complexes in the presence of the indicated variants of Cpx (all at 5 μM). Horizontal scale bar represents 20 nm. **c** Force-dependent evolution of extensions for the SNARE complex between 13 and 16 pN of force in the presence of the indicated variants of Cpx. **d** Force-dependent equilibria among the three zippered conformations. Solid lines represent the fits to a three-state equilibrium model (Supplementary Note 5). The curves for wild-type Cpx (WT) are overlaid as shades in the top panel with NAC for the sake of comparison. The color code is the same as in **c**. **e**, **f** Fold changes at 14 pN in the half-zippered (**e**) and linker-open (**f**) populations upon addition of each Cpx variant. In **d**–**f**, data are represented as mean ± s.d. of more than three measurements. **g**, **h** Representative time traces for the SNARE complexes in the presence of the indicated variants of Cpx. Vertical and horizontal scale bars represent 10 nm and 50 ms, respectively

has been largely neglected compared to the SNARE motifs. On the basis of our observation that Cpx serves as a focused clamp for the SNARE complexes in linker-open conformation, we suggest that the linker domains may represent the final molecular switch of the neuronal SNARE complex that must be zippered to accomplish full synaptic vesicle fusion[22,34,49,50] (Fig. 6, left). Indeed, a previous study with murine hippocampal neurons reported that a mutation in the linker region of synaptobrevin-2 gives rise to the loss-of-function phenotype of Cpx, suggesting that Cpx controls the force transfer from the SNARE motifs to the fusing membranes[22]. We found that the linker-open state becomes most populated above 14 pN, and that this state would separate the two fusing membranes by ~6 nm. This physical gap, largely consistent with that observed in electron microscopy studies[51,52], may act as a crucial barrier inhibiting spontaneous fusion events beyond what is functionally necessary.

We noted that the effects of mechanical tension on single SNARE complexes manifested only in a narrow range, namely between 13 and 16 pN. When the tension was below 13 pN, it did not induce appreciable conformational perturbation to the fully assembled SNAREs. Only above 13 pN, the SNARE complex began to sample intermediate conformations including the linker-open and half-zippered states. Beyond 16 pN, synaptobrevin-2 readily unraveled from the Q-SNAREs even in the presence of Cpx, and all SNAREs unfolded in turn. This observation implies that the tension above 16 pN is either not reached during synaptic vesicle fusion, not relevant to the function of neuronal SNARE complex, or both. Thus, this force range between 13 and 16 pN likely corresponds to the most important regime where multiple conformations are dynamically sampled by individual SNARE complexes.

Remarkably enough, all of the molecular effects of Cpx we observed appeared in this narrow range of tension. The function of Cpx was thus devoted to changing the balance of the conformational intermediates of SNARE complexes. In particular, the molecular actions by the distinct domains of Cpx were found to conspire to promote the linker-open conformation, making it as the only abundant state above 14 pN tension. Collectively, these results suggest that Cpx is inherently designed to work with neuronal SNARE complexes under a well-defined range of mechanical tension.

How do these observations under force-loaded environment translate to the synaptic vesicle fusion in the physiological milieu? The transitions between conformational intermediates occurred near an equilibrium condition, which rendered the relative populations of intermediates independent of force history. This hypothesis was supported by the resemblance of intermediate populations in unzipping and rezipping experiments. Consequently, an important physical condition that validates the physiological relevance of our observation is whether a single neuronal SNARE complex experiences tensions above 13 pN at synapses. Although the repulsive force between a synaptic vesicle and a presynaptic membrane has, to our knowledge, not been measured experimentally, it is predicted to quickly build up upon close apposition of the two charged membranes[31] and indeed estimated to reach 13 pN at a separation of 2–3 nm (Supplementary Fig. 8). Therefore, we suggest that the effects of Cpx described here would appear as soon as the linker domains of SNAREs attempt to zipper, providing a point for timely and effective regulation of membrane fusion. In this regard, it is remarkable that we verified the effects of Cpx during the zippering of SNARE complexes that arguably mimics the initial docking of synaptic vesicles to the membrane in neurons.

Intriguingly, we found that the two-faceted function of Cpx was attributed to distinct domains comprising Cpx. The

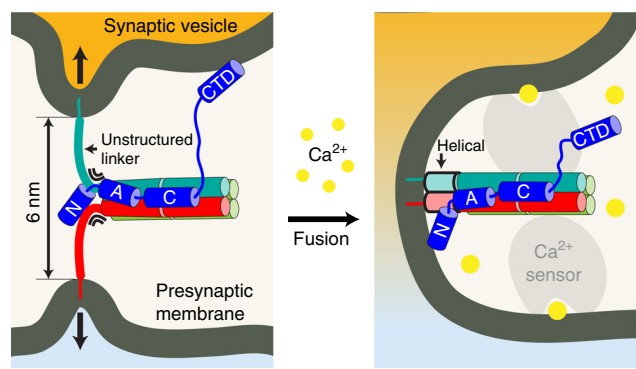

**Fig. 6** A model for the function of complexin in synaptic vesicle fusion. (Left) A synaptic vesicle tethered to a presynaptic membrane via a Cpx–SNARE complex under a force-loaded environment. Cpx is shown in blue. (Right) Cpx–SNARE complex after a Ca²⁺-triggered membrane fusion event. A putative Ca²⁺ sensor in collaboration with the Cpx–SNARE complex is shown in gray

stabilizing effect was mostly mediated by the central and accessory helices of Cpx that buttressed the four-helix bundle of SNARE motifs. The clamping function, on the contrary, was unexpectedly mediated by the N-terminal domain that interfered with the zippering of SNARE linker domains. This result is rather surprising given that the N-terminal region of Cpx is hypothesized to play a stimulatory role in triggered neurotransmission[17,23,24]. We note that our experiments were performed in the absence of phospholipid membranes, thereby precluding observation of any membrane-associated effects. We suggest a possibility that the clamping and stimulatory functions of the N-terminal domain, rather than competing with each another, manifest in a sequential manner along the steps of synaptic vesicle fusion. In the primed, linker-open state, the N-terminal domain of Cpx would be placed ~3-nm away from the fusing membranes and preferentially interact with the linker domains of SNAREs. During fusion, SNAREs will zipper further, which then allows the interaction between membrane and Cpx and liberates the stimulatory effects of Cpx[53].

We speculate on two potential mechanisms by which the final step of fusion could be triggered by a Ca²⁺ influx (Fig. 6, right). In one scenario, Ca²⁺-responsive factors such as synaptotagmin[54–60] may dislodge the N-terminus of Cpx from a SNARE complex. This release allows the linker domains of SNAREs to zipper completely, thereby strongly driving the full zippering of SNAREs even without lowering of tension. In an alternative scenario, the Ca²⁺ sensor may bridge the two fusing membranes[61–65], effectively reducing the tension applied to each SNARE complex. If this tension is reduced below 13 pN, the SNARE complex will strongly favor the fully zipped state regardless of Cpx. Full synaptic vesicle fusion will then occur without the need for any interaction between Cpx and the Ca²⁺-responsive factors.

In conclusion, our observations reaffirm the notion that the synaptic fusion machinery is a sophisticated mechanical system designed to accomplish membrane fusion within a millisecond while simultaneously clamping unwanted release of vesicles. Our results suggest that the interactions of neuronal SNARE complexes with their regulators must be understood in the presence of physiologically relevant levels of mechanical tension. An effective decrease in mechanical tension by only a few piconewtons, induced by fusion regulators, will unleash the zippering potential of neuronal SNAREs and drive synchronous neurotransmitter release.

## Methods

**Expression of recombinant proteins**. For the SNARE proteins used in this study, 6×His-tagged rat syntaxin-1A (191–268, I202C/K266C), rat SNAP-25 isoform b (2–206, C85A/C88A/C90A/C92A), and rat synaptobrevin-2 (2–97, L32C/I97C) were cloned into pET28a vector. Rat Cpx-1 variants (1–134, 1–80, 32–80, and 48–73) were cloned into HRV 3C cleavage site-inserted pGEX-4T-1 vector. The difference between rat and human proteins in the region we studied are (rat/human): none in syntaxin-1A and SNAP-25b; V8/A in synaptobrevin-2; V61/A, M62/V, P109/V, F132/L (the former two in the central helix and the latter two in the C-terminal domain) in Cpx-1. All proteins were expressed in *E. coli* Rosetta (DE3) pLysS strain (Novagen). For Cpx, cells were grown in Luria–Bertani broth (LB) with 100 μg ml⁻¹ ampicillin and 34 μg ml⁻¹ chloramphenicol. For expressing SNARE proteins, 25 μg ml⁻¹ kanamycin was used instead of ampicillin. Bacteria in 25 ml LB overnight culture (37 °C, 220 r.p.m.) were transferred into the main culture. Cells were grown (37 °C, 220 r.p.m.) to an optical density of 0.7–0.8 (600 nm) and 0.5 mM isopropyl-β-ᴅ-1-thiogalactoside (IPTG) was added to induce expression. After further incubation for 3–4 h (37 °C, 220 rpm), cells were centrifuged at 5000×g for 10 min Harvested cell pellets were frozen with liquid nitrogen and stored at −80 °C.

**Purification of recombinant proteins**. Buffers used for protein purification are listed in Supplementary Note 1. Cell pellets expressing recombinant proteins were thawed at room temperature, suspended in 30 ml of ice-cold lysis buffer, and broken up by sonication on ice. Lysates were centrifuged at 15,000 × g for 30 min at 4 °C to remove insoluble materials. Then, the supernatant was bound to either 1 ml of GST–agarose resin (Incospharm) or nickel agarose resin (Qiagen). Bound resins were poured into a gravity column and washed with 100 ml of wash buffer, and the bound proteins were eluted with elution buffer. For Cpxs, HRV-3C protease supplemented to cleave the target proteins from GST (4 °C, 90 min) in the elution buffer. Full-length Cpx (1–134) were loaded onto a Superdex 200 Increase column pre-equilibrated with elution buffer for further purification. The peak fractions were collected and concentrated (Amicon Ultra 10K, Merck). Other Cpx fragments were further purified and concentrated with proper molecular weight cut-off membrane filters. Purified proteins were verified on 12% and 20% SDS poly-acrylamide gels for SNARE and Cpx proteins, respectively. All proteins were frozen with liquid nitrogen and stored at −80 °C until use.

**Conjugation of SNARE complex to DNA**. Each of the two 510-bp DNA handles was produced by polymerase chain reaction using 5′-thiol-modified and either 5′-biotin-labeled or 5′-digoxigenin-labeled primers (Bionics) (Supplementary Note 2). The products were incubated with 100 mM DTT for 12 h at 37 °C to cleave unwanted disulfide bonds. After purification using QIAGEN-tip column (Qiagen), the handles were concentrated and activated with 10 mM 2,2′-dithiodipyridine (DTDP) for 12 h at 37 °C. After activation, the remaining DTDP was removed using QIAGEN-tip column and ethanol-precipitated DNA pellets were solubilized with phosphate-buffered saline (PBS, pH 7.4) at the final step. Purified handles were concentrated to 3–4 μM and stored in 4 °C. To conjugate DNA handles to the SNARE complex, each SNARE protein was first mixed at an equal molar ratio and incubated with 5 mM DTT for 3 h at 25 °C for assembly. Thrombin was added and dialyzed against PBS (containing 1 mM DTT) overnight at 4 °C to remove histidine tags. To remove excessive DTT, SNARE complexes were desalted using PBS-equilibrated PD MiniTrap G-25 column (GE Healthcare) and concentrated to 58 μM. The DTDP-activated biotin-labeled DNA handle was added to the solution of SNARE complex at a molar ratio of 1:20. The SNARE/DNA mixture was incubated at room temperature for 30 min. Then, an excessive amount of digoxigenin-labeled DNA handle was added to the mixture and incubated for 3 h at room temperature for complete attachment. Since we do not control the reaction specificity of the two DNA handles to syntaxin-1A and synaptobrevin-2, the product was a mixture of the two kinds (i.e., biotin–syntaxin–synaptobrevin–digoxigenin and biotin–synaptobrevin–syntaxin–digoxigenin) that were indistinguishable in tweezing experiments. Samples were frozen with liquid nitrogen and stored at −80 °C.

**Preparation of sample chamber for magnetic tweezers**. Sample chambers for magnetic tweezers were constructed from no. 1.5 glass coverslips[33,35]. A flow cell was assembled with two glass coverslips passivated with polyethylene glycols (PEG) (Laysan Bio), sandwiched with a double-sided tape spacer. A fraction (1–5%) of the PEG molecules were modified with biotin for the surface attachment of SNARE −DNA constructs via NeutrAvidin. Sequential injection of SNARE–DNA conjugate mixed with NeutrAvidin (Thermo Fisher), reference beads (Spherotech, 3.0–3.4 μm in diameter), and magnetic beads (Thermo Fisher, M270) coated with anti-digoxigenin (Sigma-Aldrich) into the flow cell yielded ~1-kbp bead–tether constructs. All measurements were performed in PBS (pH 7.4) supplemented with 2.5 μM SNAP-25 to allow refolding of SNARE in case of unfolding events at high forces. All injection and washing steps were performed with a syringe pump at flow rates of 25–1000 μl min⁻¹. When using Cpx, additional buffer exchange with a solution containing 5 μM Cpx was conducted. Full details of the sample assembly procedure are given in Supplementary Table 1.

**Magnetic tweezers setup and bead tracking**. The magnetic tweezers apparatus was built on an inverted microscope (Olympus). A pair of magnets (vertically aligned in opposite directions with a 1-mm gap) was placed above the stage holding a flow cell. The vertical position and rotation of the magnets were controlled by a translation stage (Physik Instrumente) and a stepper motor (Autonics), respectively. Beads in a flow cell were illuminated by a red superluminescent diode (QPhotonics) and imaged by a 100× oil-immersion objective (Olympus) and a high-speed CMOS camera (Mikrotron). The objective position was controlled by a piezo-controlled nano-positioner (Mad City Labs) to calibrate distances and to correct for mechanical/thermal drift. The acquired images were grabbed by a frame grabber (Active Silicon) and retrieved by a custom software written in LabVIEW (National Instruments). The three-dimensional coordinates of reference and magnetic beads were tracked in real time at 100–1200 Hz by a custom software written in LabVIEW[66].

**Force calibration and scaling**. Magnetic forces were measured and calibrated as described in the literature[67]. Forces were estimated from the power spectral densities for the Brownian motions of magnetic beads perpendicular to the magnetic field, correcting for near-surface viscosity and blurring and aliasing in image acquisition[68]. Forces were calibrated against the distance between magnets and imaging surface using 5.4-kbp DNA and the resulting force–magnet position data were fit with a double-exponential function[69]. For the measurements on SNARE complex, the average unzipping force measured in each pulling construct (SNARE–DNA–bead) without Cpx was normalized to the global average of the unzipping forces (14.4 pN) measured form multiple constructs (Supplementary Fig. 4), to minimize the bead-to-bead variation in force.

**Verification of SNARE complex in force-ramp experiments**. The end-to-end extension of SNARE complex was monitored in real time by tracking the bead position in z following a standard method in magnetic tweezers[66]. The sample specificity was first verified by checking the force–extension curves in force-ramp measurements in the range 0.1–20 pN at force loading rates of ±1 pN s⁻¹. Only the constructs exhibiting all of the signature events of SNARE complex (unzipping at 12–16 pN, rezipping at 8–12 pN, unfolding at 15–18 pN, and refolding under 2 pN) were subjected to further investigation. The unzipping and rezipping events were identified programmatically to obtain the distribution of transition force over many cycles. When necessary, the force–extension graph was compared to the model extension for the SNARE–DNA conjugate (Supplementary Note 3).

**Kinetic measurements in force-jump experiments**. In force-jump experiments, magnet was moved at the maximum speed (15 mm s⁻¹) to instantaneously change the applied force. In 10–16 pN range, this motion leads to >50 pN s⁻¹ changes in force, reaching target forces within 0.1 s. The unzipping and rezipping events were identified programmatically based on the abrupt large shifts in extension to measure the lifetimes of states at the target force. The resulting lifetimes were used to sketch the energy landscape for the unzipping/rezipping transition using the Bell equation (Supplementary Note 4). For example, the distance ($\Delta x^{\ddagger}_{unzip}$) and height ($\Delta G^{\ddagger}_{unzip}$) of the energy barrier from the zippered state was extracted from the exponential increase in the unzipping rate ($k_{unzip}$), with increasing applied force ($F$):

$$k_{unzip}(F) = k_w \exp\left[\left(-\Delta G^{\ddagger}_{unzip}(F) + F\Delta x^{\ddagger}_{unzip}\right)/k_B T\right], \qquad (1)$$

where $k_w$ represents the diffusion-limited transition rate for protein folding[70] ($= 1 \times 10^6$ s⁻¹) and $k_B T$ the thermal energy (4.11 pN nm). The values for $\Delta G^{\ddagger}_{unzip}$ and $\Delta x^{\ddagger}_{unzip}$ were obtained from linear fits of the measured unzipping rates to the logarithmic representation of Eq. (1) (Fig. 2e):

$$k_B T \log\left(k_{unzip}/k_w\right) = -\Delta G^{\ddagger}_{unzip} + F\Delta x^{\ddagger}_{unzip}. \qquad (2)$$

Analogous equations were used to calculate the location of the energy barrier from the zippered state.

**High-speed measurements of SNARE complex conformation**. The conformational equilibrium of SNARE complex was measured by tracking the magnetic bead at 1.2 kHz for >15 s at each force (long enough to reach a thermal equilibrium), and varying the force by 0.2 pN in the range 12–16 pN. The resulting distribution of bead coordinates was fit to a Gaussian mixture model (using fitgmdist in MATLAB (Mathworks)) with either two or three components depending on the level of force and the presence of Cpx. The locations and equilibrium populations of the intermediate states were obtained as the means and proportions of the identified Gaussian components, respectively.

**Code availability**. The custom LabVIEW and MATLAB codes used for data acquisition and analysis are available at https://github.com/tyyoonlab and upon request.

## Data availability

The data that support the findings of this study are available from the corresponding author upon reasonable request.

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

## Acknowledgements

We thank Mary Munson for critical reading of the manuscript, Yongli Zhang, Jekyung Ryu, and Duyoung Min for helpful discussions, and Janghyun Yoo and Hyunwoo Kim for the help with setting up high-speed magnetic tweezers. This work was supported by the National Creative Research Initiative Program (Center for Single-Molecule Systems Biology to T.-Y.Y.; grant number: NRF-2011-0018352) funded by the National Research Foundation of Korea. M.J.S. was supported by the BK21 Plus Program from the Korean Ministry of Education.

## Author contributions

T.-Y.Y. conceived of the project. All authors designed the experiments. H.K. purified recombinant proteins and prepared protein–DNA constructs. M.J.S. performed magnetic tweezers experiments and analyzed data. M.J.S. and T.-Y.Y. wrote the manuscript.

## Additional information

**Competing interests:** The authors declare no competing interests.

