## [Peer Review File · Nature Communications]

Reviewers' comments:

Reviewer #1 (Remarks to the Author):

See attachment

Reviewer #2 (Remarks to the Author):

In this work, the effects of complexin on disassembly and reassembly of the SNARE complex are examined using single molecule magnetic tweezers. The results suggest that the effects of complexin are especially pronounced at mechanical tensions around 13 pN. Under this condition, complexin stabilizes the central four helix bundle, but, at the same time, it also inhibits zippering at the C-terminal (membrane proximal) end of the SNARE complex. More specifically, it is suggested that the linker domains are prevented from zippering that connect the SNARE complex with the transmembrane domains of syntaxin and synaptobrevin. This conclusion is largely based on interpretation of the single molecule force and extension data by modeling. The capture of the stabilized partially splayed-open SNARE complex by complexin is an important contribution to the field since it lends further credence that this splayed-open state is part of the primed synaptic complex. I recommend publication after some minor revision.

Comments:

1. Introduction: the summary of various functions of complexin is fine, but the authors may also wish to refer to recent reviews, e.g., <https://doi.org/10.1016/j.tcb.2018.03.004>.
2. Introduction: On a more general note, is there an estimate in the literature how much mechanical force is generated by pulling membranes together (i.e., to overcome the hydration force barrier for the membranes to fuse)? Is it in the order of the 13 pN force used in the magnetic tweezer experiment?
3. Introduction: it would be nice to briefly summarize the findings of reference 37 (Choi et al., eLife, 2016) since they may have direct relation to the experiments in this work. In particular, Choi et al. observed that complexin induces a conformational change at the C-terminal end of the SNARE complex for about 50% of the single SNARE complexes, and that this effect requires the presence of the N-terminal, accessory, and core domains of complexin, similar to the requirements of the presence of these domains for the effects observed in this work.
4. Results, second paragraph. The introduction of the cysteines, and formation of disulfide bonds, also prevents formation of anti-parallel or other misfolded SNARE complex, in addition to allowing multiple rounds of interrogation. Please comment on this important feature of the experiment.
5. Figure 1c: what concentration of free SNAP-25 was needed to allow re-assembly of the

ternary SNARE complex?

6. Figure 2h suggests that in the presence of complexin, the length difference between zippered and unzipped state is increased by 2 nm. This observation might suggest that in the absence of complexin, either the zippered state is not completely zippered or that the unzipped state is not entirely unzipped. Please comment.

7. Figure 4: it is stated in the text "However, complexin did not actively induce unzipping of the assembled linker domains". What is the evidence for this conclusion? The force apparatus cannot measure events that may happen under conditions when no force is applied.

8. Discussion: It is suggested that complexin inhibits assembly of the linkers under loaded conditions. The work by Choi et al., suggests that complexin also induces a conformational change at the C-terminal end of the SNARE complex under relaxed conditions. Both effects of complexin (i.e., under loaded and under relaxed conditions) require the N-terminal, accessory, and core domains of complexin. Could these phenomena be related? Please comment.

9. Figure 6 and Discussion: Related to the previous point, Choi et al., 2016 observe substantial complexin-induced widening of the SNARE complex for FRET label pair attached to residues syntaxin 249 and synaptobrevin 82. These residues are located between layers +6 and +7 of the SNARE complex. Thus, based on the Choi et al. experiments, complexin affects the conformation of the SNARE complex at layers +6, +7, and +8, or possibly even more layers that are N-terminal of layer +6. The authors are encouraged to consider an alternative model to their "stretched-linker model (Figure 6)" where the synaptobrevin and/or syntaxin linker regions and additional adjacent residue assume a random coil conformation with a persistence length of 0.9 (Choi UB, McCann JJ, Weninger KR, Bowen ME. 2011. *Structure* 19:566–76. doi: 10.1016/j.str.2011.01.011.). Such an alternative model of a more splayed-open state may be consistent with both the author's data and the observations by Choi et al (2016). If such an alternative model is indeed plausible, perhaps figure 6 should be modified to offer both models.

Reviewer #3 (Remarks to the Author):

The manuscript by Shon et al. presents magnetic tweezers analysis of complexin mediated regulation of the SNARE system. This is a followup to the previous Nature communication paper by the same group. The paper is well written, arguments are clear and the methodology is appropriately chosen.

1- The main finding is that complexin acts as a force clamp by stabilising SNARE motif and modulation of zipping. The authors need to discuss previous reports (e.g. Anton Maximov et al. *Science* (2009)) on the role of complexin as a force clamp in details.

2- The study is based on rat proteins. It would be important to discuss the similarity and differences between human SNARE and complexin with those of rat.

3- The system does not include interactions with membrane. Internal membrane domains are removed for simplicity. Even without these internal membrane domains, the proteins may interact with the lipid headgroups. I suggest the authors to discuss the implications of membrane interactions for their finding.

This is a very interesting paper that describes a study of how complexin affects the energy landscape of the neuronal SNARE complex using magnetic tweezers. This subject is very important to understand the mechanism of neurotransmitter release because the SNARE complex is critical to trigger synaptic vesicle fusion and because complexin is a key regulator with active and inhibitory roles in release. The data described in the paper show that complexin binding considerably stabilizes the SNARE four-helix bundle, but also hinders zippering of the linker domains that connect the SNARE motifs of synaptobrevin and syntaxin-1 with their respective transmembrane regions. The study appears to be performed very carefully and has a large amount of data that will be of interest to the field. Hence, I strongly recommend publication in *Nature Communications*, although I do have some concerns that the authors should address before publication, and other concerns that they may also want to consider.

1. The most important concern is that the experiments were performed in the absence of membranes, which contain negatively charged phospholipids such as PS and PIP2 that are expected to interact with the linker domains. Hence, it is unclear whether the interactions of complexin with the linker domains that underlie the results described by the authors would occur in the native membrane environment. This concern should not prevent publication of the very interesting data obtained by the authors, but they should point out this (basically unavoidable) caveat of their experiments.

2. The observation that the N-terminal region of complexin hinders the zippering of the linker domain is interesting, but the authors should point out that this region plays a stimulatory role in release, rather than an inhibitory one. This appears to be a conundrum that will require clarification with further research outside the scope of this paper.

3. The models used by the authors for the intermediates of SNARE complex assembly generally assume that when synaptobrevin becomes unzipped the three helices formed by SNAP-25 and syntaxin-1 remain structured, forming a three helix bundle. While this feature has been observed by crystallography (ref. 25), the C-terminal half of the SNARE complex is known to become unstructured in solution when the C-terminus of synaptobrevin is truncated (ref. 10). Hence, if the authors want to make the models more realistic based on the available experimental evidence, they should modify them accordingly, although this is not critical for the interpretation of the data.

4. The authors may want to point out that the stabilization of the SNARE complex by complexin observed in their experiments correlates very well with the stabilization observed by H/D exchange experiments monitored by NMR spectroscopy (ref. 19).

5. It would be desirable if the authors briefly described in the main text how they calculated the energy barrier in Fig. 2h, leaving the details for the methods section.

6. The x-axis of Figs. 3e and 3h should be labeled with numbers and with the parameter being measured.

Josep Rizo

Point-by-point response to the comments on Shon *et al.*, “Focused clamping of a single neuronal SNARE complex by complexin under high mechanical tension”

Reviewer #1

This is a very interesting paper that describes a study of how complexin affects the energy landscape of the neuronal SNARE complex using magnetic tweezers. This subject is very important to understand the mechanism of neurotransmitter release because the SNARE complex is critical to trigger synaptic vesicle fusion and because complexin is a key regulator with active and inhibitory roles in release. The data described in the paper show that complexin binding considerably stabilizes the SNARE four-helix bundle, but also hinders zippering of the linker domains that connect the SNARE motifs of synaptobrevin and syntaxin-1 with their respective transmembrane regions. The study appears to be performed very carefully and has a large amount of data that will be of interest to the field. Hence, I strongly recommend publication in *Nature Communications*, although I do have some concerns that the authors should address before publication, and other concerns that they may also want to consider.

>> We thank the reviewer for the positive reception of our work.

1. The most important concern is that the experiments were performed in the absence of membranes, which contain negatively charged phospholipids such as PS and PIP2 that are expected to interact with the linker domains. Hence, it is unclear whether the interactions of complexin with the linker domains that underlie the results described by the authors would occur in the native membrane environment. This concern should not prevent publication of the very interesting data obtained by the authors, but they should point out this (basically unavoidable) caveat of their experiments.

>> We agree with the reviewer comment that the significance of membrane environment cannot be overlooked. Following the reviewer comment, we stated in Discussion of the revised manuscript that (p19):

“We note that our experiments were performed in the absence of phospholipid membranes, thereby precluding observation of any membrane-associated effects of Cpx.”

2. The observation that the N-terminal region of complexin hinders the zippering of the linker domain is interesting, but the authors should point out that this region plays a stimulatory role in release, rather than an inhibitory one. This appears to be a conundrum that will require clarification with further research outside the scope of this paper.

>> We understand reviewer's concern that the clamping function of the N-terminal domain observed in this work is seemingly contradictory to the previously reported, stimulatory roles of the same region. We note that the clamping function manifests when this N-terminal domain of Cpx mainly interacts with the linker domains of SNAREs while being distant from the fusing membranes by a ~3-nm gap. On the other hand, the stimulatory function would become visible when SNAREs complete the zippering process and the N-terminal domain of Cpx interacts with the membranes. In the revised manuscript, we thus suggest a possibility that these two functional roles of the N-terminal domain, rather than competing with one another, manifest in a sequential manner along the synaptic vesicle fusion steps. This hypothesis warrants further investigation, and we are making a progress in adding membrane patches in our magnetic tweezers experiments using the nanodisc system.

Regarding the comments #1 and #2, we added the following paragraph in Discussion of the revised manuscript (p19):

“Intriguingly, we found that the two-faceted function of Cpx was attributed to distinct domains comprising Cpx. The stabilizing effect was mostly mediated by the central and accessory helices of Cpx that buttressed the four-helix bundle structure of the SNARE motifs. The clamping function, on the contrary, was unexpectedly mediated by the N-terminal domain that interfered with zippering of the SNARE linker domains. This result is rather surprising given that the N-terminal region of Cpx is hypothesized to play a stimulatory role in triggered neurotransmission^{17,23,24}. We note that our experiments were performed in the absence of phospholipid membranes, thereby precluding observation of any membrane-associated effects of Cpx. We suggest a possibility that the clamping function and the stimulatory function of the N-terminal domain, rather than competing with one another, manifest in a sequential manner along the synaptic vesicle fusion steps. In the primed, linker-open state, the N-terminal domain of Cpx would be mainly interacting with the linker domains of SNAREs and displaced from the fusing membranes by a ~3-nm gap. In the actual fusion step, SNAREs will complete their zippering process, which then allows the interaction between membrane and Cpx and discharges the stimulatory effects of Cpx⁵³.”

3. The models used by the authors for the intermediates of SNARE complex assembly generally assume that when synaptobrevin becomes unzipped the three helices formed by SNAP-25 and syntaxin-1 remain structured, forming a three helix bundle. While this feature has been observed by crystallography (ref. 25), the C-terminal half of the SNARE complex is known to become unstructured in solution when the C-terminus of synaptobrevin is truncated (ref. 10). Hence, if the authors want to make the models more realistic based on the available experimental evidence, they should modify them accordingly, although this is not critical for the interpretation of the data.

>> We thank the reviewer for this insightful comment. We re-examined our theoretical modeling for the intermediate conformations of the SNARE complex, and indeed found that in the half-zipped state (i.e., half-unzipped state), the Q-SNAREs also undergo partial unfolding up to the layer +4 (synaptobrevin is unraveled up to the layer +2). In the fully unzipped state (for synaptobrevin), this partial unfolding of Q-SNAREs is further increased to the zeroth layer. In the revised manuscript, we updated these calculated extension values and also revised our descriptions of the model on p11 (and in **Supplementary Note**, “Calculation of extension for the SNARE–DNA conjugate”):

“We assigned the lowest peak to the fully zippered SNARE complex and defined it as 0 nm for simplicity. The next two peaks, located at 5.3 nm and 13.0 nm, matched the expected lengths for the linker-open (5.4 nm) and the half-zipped state (13.9 nm), respectively (**Fig. 3b** versus **3a**; see **Supplementary Notes** for calculations of the expected lengths). For the linker-open conformation, we assumed fully folded four-helix bundles of the SNARE motifs and random-coil linker domains void of secondary structures (**Fig. 3b**)^{34,45}. For the half-zipped state, we included unzipping of synaptobrevin-2 and the Q-SNARE proteins up to the +2 and +4 layers, respectively^{11,37,46}. The peak after the main unzipping was placed at 28.5 nm. This observed extension best correlated with complete unraveling of synaptobrevin-2 and increased unfolding of the Q-SNARE proteins to the zeroth layer, which had an estimated extension of 26.7 nm (**Fig. 3b** versus **3a** and **Supplementary Fig. 3**).”

4. The authors may want to point out that the stabilization of the SNARE complex by complexin observed in their experiments correlates very well with the stabilization observed by H/D exchange experiments monitored by NMR spectroscopy (ref. 19).

>> The reviewer comment was well taken, and we added this point in Discussion (p17).

“Using single-molecule magnetic tweezers, we observed that the Cpx binding significantly enhances the mechanical stability of single SNARE complexes. Our observation is consistent

with the previous deuterium exchange data that Cpx physically protects the four-helix bundle structure of the SNARE motifs²⁰.”

5. It would be desirable if the authors briefly described how they calculated the energy barrier in Fig. 2h, leaving the details for the methods section.

>> We added a brief description for how we calculated the parameter of the energy barrier in **Fig. 2h**. We also added more details of calculating the energy barrier parameters in the Methods section:

Results (p9): “From the linear dependences of the logarithms of the kinetic rates on the applied mechanical tension (**Fig. 2e,g**), we deduced distances and heights of the energy barriers crossed during unzipping and re-zipping of a single neuronal SNARE complex. We present a coarse-grained sketch of this energy landscape in the presence or absence of Cpx (**Fig. 2h**; we used a pre-exponential factor of 10^6 s^{-1} as generally accepted^{33,34}).”

Methods (p25): “The resulting lifetimes were used to sketch the energy landscape for the unzipping/re-zipping transition using the Bell equation (**Supplementary Notes**). For example, the distance ($\Delta x_{\text{unzip}}^\ddagger$) and height ($\Delta G_{\text{unzip}}^\ddagger$) of the energy barrier from the zippered state was extracted from the exponential increase in the unzipping rate (k_{unzip}), with increasing applied force (F):

$$k_{\text{unzip}}(F) = k_w \exp\left[\frac{-\Delta G_{\text{unzip}}^\ddagger(F) + F\Delta x_{\text{unzip}}^\ddagger}{k_B T}\right], \quad (1)$$

where k_w represents the diffusion-limited transition rate for protein folding⁷⁰ ($= 1 \times 10^6 \text{ s}^{-1}$) and $k_B T$ the thermal energy (4.11 pN·nm). The values for $\Delta G_{\text{unzip}}^\ddagger$ and $\Delta x_{\text{unzip}}^\ddagger$ were obtained from linear fits of the measured unzipping rates to the logarithmic representation of **Eq. 1 (Fig. 2e)**:

$$k_B T \log(k_{\text{unzip}}/k_w) = -\Delta G_{\text{unzip}}^\ddagger + F\Delta x_{\text{unzip}}^\ddagger. \quad (2)$$

Analogous equations were used to calculate the location of the energy barrier from the zippered state.”

6. The x-axis of Figs. 3e and 3h should be labeled with numbers and with the parameter being measured.

>> We added the numbers accordingly.

Reviewer #2

In this work, the effects of complexin on disassembly and reassembly of the SNARE complex are examined using single molecule magnetic tweezers. The results suggest that the effects of complexin are especially pronounced at mechanical tensions around 13 pN. Under this condition, complexin stabilizes the central four helix bundle, but, at the same time, it also inhibits zippering at the C-terminal (membrane proximal) end of the SNARE complex. More specifically, it is suggested that the linker domains are prevented from zippering that connect the SNARE complex with the transmembrane domains of syntaxin and synaptobrevin. This conclusion is largely based on interpretation of the single molecule force and extension data by modeling. The capture of the stabilized partially splayed-open SNARE complex by complexin is an important contribution to the field since it lends further credence that this splayed-open state is part of the primed synaptic complex. I recommend publication after some minor revision.

>> We thank the reviewer for the positive evaluation of our work.

1. Introduction: the summary of various functions of complexin is fine, but the authors may also wish to refer to recent reviews, e.g., <https://doi.org/10.1016/j.tcb.2018.03.004>.

>> As suggested, we cited two recent reviews on the function of complexin in the synaptic vesicle fusion.

5. Trimbuch, T. & Rosenmund, C. Should I stop or should I go? The role of complexin in neurotransmitter release. *Nat. Rev. Neurosci.* **17**, 118–125 (2016).
6. Brunger, A. T., Leitz, J., Zhou, Q., Choi, U. B. & Lai, Y. Ca²⁺-Triggered Synaptic Vesicle Fusion Initiated by Release of Inhibition. *Trends Cell Biol.* (2018). doi:10.1016/j.tcb.2018.03.004

2. Introduction: On a more general note, is there an estimate in the literature how much mechanical force is generated by pulling membranes together (i.e., to overcome the hydration force barrier for the membranes to fuse)? Is it in the order of the 13 pN force used in the magnetic tweezer experiment?

>> We suppose that the electrostatic repulsion arises when synaptic vesicles initially approach the plasma membrane, which was estimated to reach 13 pN near a 2-nm separation (**Supplementary Fig. S8** and Bykhovskaia *et al.*, *Biophys. J.* **105**, 679 (2013)). Hydration barrier can certainly generate much higher repulsion for closely apposed membranes (Oelkers *et al.*, *PNAS* **113**, 13051 (2016)). In Introduction, we included hydration barrier and steric hindrance when discussing the causes for the repulsion between fusing membranes (p4), and added a reference paper that studied the hydration barrier.

“It has been presumed that the tension rapidly builds up when a synaptic vesicle approaches a presynaptic membrane because of the electrostatic repulsion, hydration barrier, and steric hindrance between the two fusing membranes^{31,32}. This tension in a SNARE complex was shown to significantly tilt the energy landscape that governs SNARE zipper processes^{33,34}.”

32. Oelkers, M., Witt, H., Halder, P., Jahn, R. & Janshoff, A. SNARE-mediated membrane fusion trajectories derived from force-clamp experiments. *Proc. Natl. Acad. Sci. U.S.A.* **113**, 13051–13056 (2016).

3. Introduction: it would be nice to briefly summarize the findings of reference 37 (Choi et al., eLife, 2016) since they may have direct relation to the experiments in this work. In particular, Choi et al. observed that complexin induces a conformational change at the C-terminal end of the SNARE complex for about 50% of the single SNARE complexes, and that this effect requires the presence of the N-terminal, accessory, and core domains of complexin, similar to the requirements of the presence of these domains for the effects observed in this work.

>> We will address this point together with the related comments #7 and #8.

4. Results, second paragraph. The introduction of the cysteines, and formation of disulfide bonds, also prevents formation of anti-parallel or other misfolded SNARE complex, in addition to allowing multiple rounds of interrogation. Please comment on this important feature of the experiment.

>> We thank the reviewer for pointing this out. We added a comment (p6):

“To enable multiple cycles of interrogation, we covalently linked the N-termini of synaptobrevin-2 and syntaxin-1A by introducing two additional cysteines that formed a disulfide bond (**Fig. 1b**). This N-terminal knotting also allowed study of only properly folded SNARE complexes during our tweezing experiments. Misfolded SNARE complexes such as anti-parallel SNARE complexes failed to form the disulfide crosslinking, ruptured under high mechanical tension, and were subsequently excluded from our observations.”

5. Figure 1c: what concentration of free SNAP-25 was needed to allow re-assembly of the ternary SNARE complex?

>> Typically 2.5 μM , but we find that as little as 1 μM was sufficient to allow refolding within tens of seconds. We put this number previously on p23 and **Supplementary Table 1**, and now on p7, too.

6. Figure 2h suggests that in the presence of complexin, the length difference between zippered and unzipped state is increased by 2 nm. This observation might suggest that in the absence of complexin, either the zippered state is not completely zippered or that the unzipped state is not entirely unzipped. Please comment.

>> The zippered state presented in **Fig. 2h** refers to the coarse-grained collection of fully and partially zippered intermediates shown in **Fig. 3**. Therefore, we think the shift in the zippered state simply reflects an equilibrium shift in the zippered intermediates (toward left, “more zippering”), consistent with a significant loss of the half-zippered state in the presence of Cpx.

7. Figure 4: it is stated in the text "However, complexin did not actively induce unzipping of the assembled linker domains". What is the evidence for this conclusion? The force apparatus cannot measure events that may happen under conditions when no force is applied.

8. Discussion: It is suggested that complexin inhibits assembly of the linkers under loaded conditions. The work by Choi et al., suggests that complexin also induces a conformational change at the C-terminal end of the SNARE complex under relaxed conditions. Both effects of complexin (i.e., under loaded and under relaxed conditions) require the N-terminal, accessory, and core domains of complexin. Could these phenomena be related? Please comment.

>> For the comments #3, #7 and #8, the reviewer is correct in pointing out that under high mechanical tension above 13 pN, the transition from the fully zippered to the linker-open state was unaffected by Cpx (**Fig. 4e**), which supports the notion of passive inhibition of linker-domain assembly by Cpx. Inspired by reviewer’s points and the results in Choi et al., we re-examined our data and generated a new plot (**Supplementary Fig. 7a**) that might bridge the observations in high- and low-force regimes. Interestingly, extrapolation of the data in **Fig. 4e** toward 0 pN indicates a faster opening of the linker domains in the presence of Cpx by an order of magnitude ($2 \times 10^{-3} \text{ s}^{-1}$ with 5 μM wild-type Cpx versus $2 \times 10^{-4} \text{ s}^{-1}$ without Cpx at 0 pN). This aspect likely explains the Cpx-induced conformational changes reported in Choi *et al.*, a conclusion from equilibrium measurements under zero force.

Supplementary Fig. 7a

We revised our description of the kinetic results as below (p14):

“Consistent with the observed promotion of the linker-open state, Cpx decreased the zippering rate of the linker domains (i.e., transition rate from the linker-open to the fully zippered state) (Fig. 4d). However, the unzipping rate of the linker domains was only marginally affected by Cpx (Fig. 4e), suggesting a rather passive role of Cpx in inhibition of the linker domain assembly. We noted that in the high-force regime we studied, the unzipping effect of the mechanical tension outcompeted any unzipping effect of Cpx. On the contrary, when extrapolating the data in Fig. 4e towards 0 pN, we found that the unzipping kinetic rate would become substantially higher in the presence of Cpx (Supplementary Fig. 7a). This analysis suggested that in such low-pN force regime, Cpx would have an effect of actively opening the linker domains (albeit on much longer time scales). This is consistent with a previous single-molecule fluorescence resonance energy transfer (FRET) study that reported lowering of the FRET efficiency at the C-terminal end of the SNARE complex by Cpx⁴⁰.”

40. Choi, U. B., Zhao, M., Zhang, Y., Lai, Y. & Brunger, A. T. Complexin induces a conformational change at the membrane-proximal C-terminal end of the SNARE complex. *Elife* **5**, (2016).

9. Figure 6 and Discussion: Related to the previous point, Choi et al., 2016 observe substantial complexin-induced widening of the SNARE complex for FRET label pair attached to residues syntaxin 249 and synaptobrevin 82. These residues are located between layers +6 and +7 of the SNARE complex. Thus, based on the Choi et al. experiments, complexin affects the conformation of the SNARE complex at layers +6, +7, and +8, or possibly even more layers that are N-terminal of layer +6. The authors are encouraged to consider an alternative model to their "stretched-linker model (Figure 6)" where the

synaptobrevin and/or syntaxin linker regions and additional adjacent residue assume a random coil conformation with a persistence length of 0.9 (Choi UB, McCann JJ, Weninger KR, Bowen ME. 2011. *Structure* 19:566–76. doi: 10.1016/j.str.2011.01.011.). Such an alternative model of a more splayed-open state may be consistent with both the author’s data and the observations by Choi et al (2016). If such an alternative model is indeed plausible, perhaps figure 6 should be modified to offer both models.

>> We modeled the linker region of SNAREs as a worm-like chain with a persistence length 0.6 nm (see **Supplementary Note**, “Calculation of extension for the SNARE–DNA conjugate”), as typically done for polypeptides in tweezers (Gao et al. *Science* 337, 1340–1343 (2012)). According to this model, unzipping of SNAREs up to layer +8 leads to an increase in extension by 5.4 nm at 14 pN (**Supplementary Fig. 6**) that consistently agrees with our data (**Fig. 3b** vs. **3a**). This increase in extension (5.4 nm) means that the linker regions would be very stretched to 64% of its contour length ($0.365 \text{ nm/AA} \times 23 \text{ AA} = 8.4 \text{ nm}$), so the linker regions in **Fig. 6** are drawn rather stretched than coiled to underscore the presence of tension.

Following your suggestion, we calculated the linker-domain extension at 14 pN using 0.9 nm for the persistence length. The results are: 6.0 nm (for the unzipping up to layer +8); 7.5 nm (layer +7); 9.6 nm (layer +6). These values do not seem very compatible with the measured distances between the observed states in **Fig. 3**, so we decided to maintain our view. We clarified that the linker region would assume a random-coil conformation (p11) and added the suggested reference:

45. Choi, U. B., McCann, J. J., Weninger, K. R. & Bowen, M. E. Beyond the random coil: stochastic conformational switching in intrinsically disordered proteins. *Structure* **19**, 566–576 (2011).

Reviewer #3

The manuscript by Shon et al. presents magnetic tweezers analysis of complexin mediated regulation of the SNARE system. This is a followup to the previous Nature communication paper by the same group. The paper is well written, arguments are clear and the methodology is appropriately chosen.

>> We thank the reviewer for the positive comment on our work.

1- The main finding is that complexin acts as a force clamp by stabilising SNARE motif and modulation of zipping. The authors need to discuss previous reports (e.g. Anton Maximov et al. Science (2009)) on the role of complexin as a force clamp in details.

>> We added this point in Discussion and also cited the mentioned paper (p17).

“On the basis of our observation that Cpx serves as a focused clamp for the SNARE complexes in the linker-open conformation, we suggest that the linker domains may represent the final molecular switch of the neuronal SNARE complex that must be zippered to accomplish full synaptic vesicle fusion^{22,34,49,50} (Fig. 6, left). Indeed, a previous study with murine hippocampal neurons reported that a mutation in the linker region of synaptobrevin-2 gives rise to the loss-of-function phenotype of Cpx, suggesting that Cpx controls the force transfer from the SNARE motifs to the fusing membranes²².”

22. Maximov, A., Tang, J., Yang, X., Pang, Z. P. & Südhof, T. C. Complexin controls the force transfer from SNARE complexes to membranes in fusion. *Science* **323**, 516–521 (2009).

2- The study is based on rat proteins. It would be important to discuss the similarity and differences between human SNARE and complexin with those of rat.

>> The rat SNARE and complexin proteins were almost identical to the human proteins in their sequence. In particular, for the regions we used, only five residues are different for the entire SNARE-Cpx system. We summarized the difference in Methods and Discussion as follows.

Methods (p21): “The difference between rat and human proteins in the region we studied are (rat/human): none in syntaxin-1A and SNAP-25b; V8/A in synaptobrevin-2; V61/A, M62/V, P109/V, F132/L (the former two in the central helix and the latter two in the C-terminal domain) in complexin-1.”

Discussion (p17): “Although our observations are based on Cpx and SNARE proteins from *Rattus norvegicus*, the results are expected to be essentially conserved for human proteins because only five residues are different when comparing the human and rat Cpx-SNARE systems (see Methods).”

3- The system does not include interactions with membrane. Internal membrane domains are removed for simplicity. Even without these internal membrane domains, the proteins may interact with the lipid headgroups. I suggest the authors to discuss the implications of membrane interactions for their finding.

>> We agree with the reviewer comment that the significance of membrane environment cannot be overlooked. Following the reviewer comment, we stated in Discussion of the revised manuscript that:

“We note that our experiments were performed in the absence of phospholipid membranes, thereby precluding observation of any membrane-associated effects of Cpx”.

Also, the clamping function of the N-terminal domain observed in this work is seemingly contradictory to the previously reported, stimulatory roles of the same region. We however note that the clamping function manifests when this N-terminal domain of Cpx mainly interacts with the linker domains of SNAREs while being distant from the fusing membranes by a ~3-nm gap. On the contrary, the stimulatory function would become visible when SNAREs complete the zippering process and the N-terminal domain of Cpx interacts with the membranes. In the revised manuscript, we thus suggest a possibility that these two functional roles of the N-terminal domain, rather than competing with one another, manifest in a sequential manner along the synaptic vesicle fusion steps. This hypothesis warrants further investigation, and we are making a progress in adding membrane patches in our magnetic tweezers experiments using the nanodisc system.

We added the following paragraph in Discussion of the revised manuscript (p19):

“Intriguingly, we found that the two-faceted function of Cpx was attributed to distinct domains comprising Cpx. The stabilizing effect was mostly mediated by the central and accessory helices of Cpx that buttressed the four-helix bundle structure of the SNARE motifs. The clamping function, on the contrary, was unexpectedly mediated by the N-terminal domain that interfered with zippering of the SNARE linker domains. This result is rather surprising given that the N-terminal region of Cpx is hypothesized to play a stimulatory role in triggered neurotransmission^{17,23,24}. We note that our experiments were performed in the absence of phospholipid membranes, thereby precluding observation of any membrane-associated effects of Cpx. We suggest a possibility that the clamping function and the stimulatory function of the N-

terminal domain, rather than competing with one another, manifest in a sequential manner along the synaptic vesicle fusion steps. In the primed, linker-open state, the N-terminal domain of Cpx would be mainly interacting with the linker domains of SNAREs and displaced from the fusing membranes by a ~3-nm gap. In the actual fusion step, SNAREs will complete their zippering process, which then allows the interaction between membrane and Cpx and discharges the stimulatory effects of Cpx⁵³.”

REVIEWERS' COMMENTS:

Reviewer #1 (Remarks to the Author):

The authors have satisfactorily addressed my concerns and I recommend publication of the paper in its current form.

Reviewer #2 (Remarks to the Author):

I would like to thank the authors for addressing my concerns and suggestions. I have no further comments, and recommend publication.

Reviewer #3 (Remarks to the Author):

The authors properly addressed my comments. I recommend publication of the revised paper at Nature Communications.